# Active Learning and Best-Response Dynamics

**Maria-Florina Balcan**
Carnegie Mellon
ninamf@cs.cmu.edu

**Christopher Berlind**
Georgia Tech
cberlind@gatech.edu

**Avrim Blum**
Carnegie Mellon
avrim@cs.cmu.edu

**Emma Cohen**
Georgia Tech
ecohen@gatech.edu

**Kaushik Patnaik**
Georgia Tech
kpatnaik3@gatech.edu

**Le Song**
Georgia Tech
lsong@cc.gatech.edu

## Abstract

We examine an important setting for engineered systems in which low-power distributed sensors are each making highly noisy measurements of some unknown target function. A center wants to accurately learn this function by querying a small number of sensors, which ordinarily would be impossible due to the high noise rate. The question we address is whether local communication among sensors, together with natural best-response dynamics in an appropriately-defined game, can denoise the system without destroying the true signal and allow the center to succeed from only a small number of active queries. By using techniques from game theory and empirical processes, we prove positive (and negative) results on the denoising power of several natural dynamics. We then show experimentally that when combined with recent agnostic active learning algorithms, this process can achieve low error from very few queries, performing substantially better than active or passive learning without these denoising dynamics as well as passive learning with denoising.

## 1 Introduction

Active learning has been the subject of significant theoretical and experimental study in machine learning, due to its potential to greatly reduce the amount of labeling effort needed to learn a given target function. However, to date, such work has focused only on the single-agent low-noise setting, with a learning algorithm obtaining labels from a single, nearly-perfect labeling entity. In large part this is because the effectiveness of active learning is known to quickly degrade as noise rates become high [5]. In this work, we introduce and analyze a novel setting where label information is held by highly-noisy low-power agents (such as sensors or micro-robots). We show how by first using simple game-theoretic dynamics among the agents we can quickly approximately denoise the system. This allows us to exploit the power of active learning (especially, recent advances in agnostic active learning), leading to efficient learning from only a small number of expensive queries.

We specifically examine an important setting relevant to many engineered systems where we have a large number of low-power agents (e.g., sensors). These agents are each measuring some quantity, such as whether there is a high or low concentration of a dangerous chemical at their location, but they are assumed to be highly noisy. We also have a center, far away from the region being monitored, which has the ability to query these agents to determine their state. Viewing the agents as examples, and their states as noisy labels, the goal of the center is to learn a good approximation to the true target function (e.g., the true boundary of the high-concentration region for the chemical being monitored) from a small number of label queries. However, because of the high noise rate, learning this function directly would require a very large number of queries to be made (for noise rate $\eta$, one would necessarily require $\Omega(\frac{1}{(1/2-\eta)^2})$ queries [4]). The question we address in this

paper is to what extent this difficulty can be alleviated by providing the agents the ability to engage in a small amount of local communication among themselves.

What we show is that by using local communication and applying simple robust state-changing rules such as following natural game-theoretic dynamics, randomly distributed agents can modify their state in a way that greatly de-noises the system without destroying the true target boundary. This then nicely meshes with recent advances in agnostic active learning [1], allowing for the center to learn a good approximation to the target function from a small number of queries to the agents. In particular, in addition to proving theoretical guarantees on the denoising power of game-theoretic agent dynamics, we also show experimentally that a version of the agnostic active learning algorithm of [1], when combined with these dynamics, indeed is able to achieve low error from a small number of queries, outperforming active and passive learning algorithms without the best-response denoising step, as well as outperforming passive learning algorithms with denoising. More broadly, engineered systems such as sensor networks are especially well-suited to active learning because components may be able to communicate among themselves to reduce noise, and the designer has some control over how they are distributed and so assumptions such as a uniform or other "nice" distribution on data are reasonable. We focus in this work primarily on the natural case of linear separator decision boundaries but many of our results extend directly to more general decision boundaries as well.

## 1.1 Related Work

There has been significant work in active learning (e.g., see [11, 15]) including active learning in the presence of noise [9, 4, 1], yet it is known active learning can provide significant benefits in low noise scenarios only [5]. There has also been extensive work analyzing the performance of simple dynamics in consensus games [6, 8, 14, 13, 3, 2]. However this work has focused on getting to *some* equilibria or states of *low social cost*, while we are primarily interested in getting near a specific desired configuration, which as we show below is an approximate equilibrium.

## 2 Setup

We assume we have a large number $N$ of agents (e.g., sensors) distributed uniformly at random in a geometric region, which for concreteness we consider to be the unit ball in $R^d$. There is an unknown linear separator such that in the initial state, each sensor on the positive side of this separator is positive independently with probability $\geq 1-\eta$, and each on the negative side is negative independently with probability $\geq 1 - \eta$. The quantity $\eta < 1/2$ is the *noise rate*.

### 2.1 The basic sensor consensus game

The sensors will denoise themselves by viewing themselves as players in a certain consensus game, and performing a simple dynamics in this game leading towards a specific $\epsilon$-equilibrium.

Specifically, the game is defined as follows, and is parameterized by a communication radius $r$, which should be thought of as small. Consider a graph where the sensors are vertices, and any two sensors within distance $r$ are connected by an edge. Each sensor is in one of two states, positive or negative. The *payoff* a sensor receives is its correlation with its neighbors: the fraction of neighbors in the same state as it minus the fraction in the opposite state. So, if a sensor is in the same state as all its neighbors then its payoff is 1, if it is in the opposite state of all its neighbors then its payoff is $-1$, and if sensors are in uniformly random states then the expected payoff is 0. Note that the states of highest social welfare (highest sum of utilities) are the all-positive and all-negative states, which are *not* what we are looking for. Instead, we want sensors to approach a different near-equilibrium state in which (most of) those on the positive side of the target separator are positive and (most of) those on the negative side of the target separator are negative. For this reason, we need to be particularly careful with the specific dynamics followed by the sensors.

We begin with a simple lemma that for sufficiently large $N$, the target function (i.e., all sensors on the positive side of the target separator in the positive state and the rest in the negative state) is an $\epsilon$-equilibrium, in that no sensor has more than $\epsilon$ incentive to deviate.

**Lemma 1** *For any $\epsilon, \delta > 0$, for sufficiently large $N$, with probability $1 - \delta$ the target function is an $\epsilon$-equilibrium.*

PROOF SKETCH: The target function fails to be an $\epsilon$-equilibrium iff there exists a sensor for which more than an $\epsilon/2$ fraction of its neighbors lie on the opposite side of the separator. Fix one sensor

$x$ and consider the probability this occurs to $x$, over the random placement of the $N - 1$ other sensors. Since the probability mass of the $r$-ball around $x$ is at least $(r/2)^d$ (see discussion in proof of Theorem 2), so long as $N - 1 \geq (2/r)^d \cdot \max[8, \frac{4}{\epsilon^2}] \ln(\frac{2N}{\delta})$, with probability $1 - \frac{\delta}{2N}$, point $x$ will have $m_x \geq \frac{2}{\epsilon^2} \ln(\frac{2N}{\delta})$ neighbors (by Chernoff bounds), each of which is at least as likely to be on $x$'s side of the target as on the other side. Thus, by Hoeffding bounds, the probability that more than a $\frac{1}{2} + \frac{\epsilon}{2}$ fraction lie on the wrong side is at most $\frac{\delta}{2N} + \frac{\delta}{2N} = \frac{\delta}{N}$. The result then follows by union bound over all $N$ sensors. For a bit tighter argument and a concrete bound on $N$, see the proof of Theorem 2 which essentially has this as a special case. ∎

Lemma 1 motivates the use of best-response dynamics for denoising. Specifically, we consider a dynamics in which each sensor switches to the majority vote of all the other sensors in its neighborhood. We analyze below the denoising power of this dynamics under both synchronous and asynchronous update models. In supplementary material, we also consider more robust (though less practical) dynamics in which sensors perform more involved computations over their neighborhoods.

## 3 Analysis of the denoising dynamics

### 3.1 Simultaneous-move dynamics

We start by providing a positive theoretical guarantee for one-round simultaneous move dynamics. We will use the following standard concentration bound:

**Theorem 1 (Bernstein, 1924)** *Let* $X = \sum_{i=1}^N X_i$ *be a sum of independent random variables such that* $|X_i - \mathrm{E}[X_i]| \leq M$ *for all* $i$. *Then for any* $t > 0$, $\mathbb{P}[X - \mathrm{E}[X] > t] \leq \exp\left(\frac{-t^2}{2(\mathrm{Var}[X] + Mt/3)}\right)$.

**Theorem 2** *If* $N \geq \frac{2}{(r/2)^d(\frac{1}{2} - \eta)^2} \ln\left(\frac{1}{(r/2)^d(\frac{1}{2} - \eta)^2 \delta}\right) + 1$ *then, with probability* $\geq 1 - \delta$, *after one synchronous consensus update every sensor at distance* $\geq r$ *from the separator has the correct label.*

Note that since a band of width $2r$ about a linear separator has probability mass $O(r\sqrt{d})$, Theorem 2 implies that with high probability one synchronous update denoises all but an $O(r\sqrt{d})$ fraction of the sensors. In fact, Theorem 2 does not require the separator to be linear, and so this conclusion applies to any decision boundary with similar surface area, such as an intersection of a constant number of halfspaces or a decision surface of bounded curvature.

**Proof (Theorem 2):** Fix a point $x$ in the sample at distance $\geq r$ from the separator and consider the ball of radius $r$ centered at $x$. Let $n_+$ be the number of correctly labeled points within the ball and $n_-$ be the number of incorrectly labeled points within the ball. Now consider the random variable $\Delta = n_- - n_+$. Denoising $x$ can give it the incorrect label only if $\Delta \geq 0$, so we would like to bound the probability that this happens. We can express $\Delta$ as the sum of $N - 1$ independent random variables $\Delta_i$ taking on value 0 for points outside the ball around $x$, 1 for incorrectly labeled points inside the ball, or $-1$ for correct labels inside the ball. Let $V$ be the measure of the ball centered at $x$ (which may be less than $r^d$ if $x$ is near the boundary of the unit ball). Then since the ball lies entirely on one side of the separator we have

$$\mathrm{E}[\Delta_i] = (1 - V) \cdot 0 + V\eta - V(1 - \eta) = -V(1 - 2\eta).$$

Since $|\Delta_i| \leq 1$ we can take $M = 2$ in Bernstein's theorem. We can also calculate that $\mathrm{Var}[\Delta_i] \leq \mathrm{E}[\Delta_i^2] = V$. Thus the probability that the point $x$ is updated incorrectly is

$$\mathbb{P}\left[\sum_{i=1}^{N-1} \Delta_i \geq 0\right] = \mathbb{P}\left[\sum_{i=1}^{N-1} \Delta_i - \mathrm{E}\left[\sum_{i=1}^{N-1} \Delta_i\right] \geq (N-1)V(1 - 2\eta)\right]$$

$$\leq \exp\left(\frac{-(N-1)^2 V^2 (1 - 2\eta)^2}{2\big((N-1)V + 2(N-1)V(1 - 2\eta)/3\big)}\right)$$

$$\leq \exp\left(\frac{-(N-1)V(1 - 2\eta)^2}{2 + 4(1 - 2\eta)/3}\right)$$

$$\leq \exp\left(-(N-1)V(\tfrac{1}{2} - \eta)^2\right)$$

$$\leq \exp\left(-(N-1)(r/2)^d(\tfrac{1}{2} - \eta)^2\right),$$

where in the last step we lower bound the measure $V$ of the ball around $r$ by the measure of the sphere of radius $r/2$ inscribed in its intersection with the unit ball. Taking a union bound over all $N$ points, it suffices to have $e^{-(N-1)(r/2)^d(\frac{1}{2}-\eta)^2} \leq \delta/N$, or equivalently

$$N - 1 \geq \frac{1}{(r/2)^d(\frac{1}{2}-\eta)^2}\left(\ln N + \ln\frac{1}{\delta}\right).$$

Using the fact that $\ln x \leq \alpha x - \ln \alpha - 1$ for all $x, \alpha > 0$ yields the claimed bound on $N$. ∎

We can now combine this result with the efficient agnostic active learning algorithm of [1]. In particular, applying the most recent analysis of [10, 16] of the algorithm of [1], we get the following bound on the number of queries needed to efficiently learn to accuracy $1 - \epsilon$ with probability $1 - \delta$.

**Corollary 1** *There exists constant $c_1 > 0$ such that for $r \leq \epsilon/(c_1\sqrt{d})$, and $N$ satisfying the bound of Theorem 2, if sensors are each initially in agreement with the target linear separator independently with probability at least $1-\eta$, then one round of best-response dynamics is sufficient such that the agnostic active learning algorithm of [1] will efficiently learn to error $\epsilon$ using only $O(d\log 1/\epsilon)$ queries to sensors.*

In Section 5 we implement this algorithm and show that experimentally it learns a low-error decision rule even in cases where the initial value of $\eta$ is quite high.

### 3.2 A negative result for arbitrary-order asynchronous dynamics

We contrast the above positive result with a negative result for arbitrary-order asynchronous moves. In particular, we show that for any $d \geq 1$, for sufficiently large $N$, with high probability there exists an update order that will cause all sensors to become negative.

**Theorem 3** *For some absolute constant $c > 0$, if $r \leq 1/2$ and sensors begin with noise rate $\eta$, and*

$$N \geq \frac{16}{(cr)^d\phi^2}\left(\ln\frac{8}{(cr)^d\phi^2} + \ln\frac{1}{\delta}\right),$$

*where $\phi = \phi(\eta) = \min(\eta, \frac{1}{2} - \eta)$, then with probability at least $1 - \delta$ there exists an ordering of the agents so that asynchronous updates in this order cause all points to have the same label.*

PROOF SKETCH: Consider the case $d = 1$ and a target function $x > 0$. Each subinterval of $[-1, 1]$ of width $r$ has probability mass $r/2$, and let $m = rN/2$ be the expected number of points within such an interval. The given value of $N$ is sufficiently large that with high probability, all such intervals in the initial state have both a positive count and a negative count that are within $\pm\frac{\phi}{4}m$ of their expectations. This implies that if sensors update left-to-right, initially all sensors will (correctly) flip to negative, because their neighborhoods have more negative points than positive points. But then when the "wave" of sensors reaches the positive region, they will continue (incorrectly) flipping to negative because the at least $m(1 - \frac{\phi}{2})$ negative points in the left-half of their neighborhood will outweigh the at most $(1 - \eta + \frac{\phi}{4})m$ positive points in the right-half of their neighborhood. For a detailed proof and the case of general $d > 1$, see supplementary material. ∎

### 3.3 Random order dynamics

While Theorem 3 shows that there *exist* bad orderings for asynchronous dynamics, we now show that we can get positive theoretical guarantees for *random order* best-response dynamics.

The high level idea of the analysis is to partition the sensors into three sets: those that are within distance $r$ of the target separator, those at distance between $r$ and $2r$ from the target separator, and then all the rest. For those at distance $< r$ from the separator we will make no guarantees: they might update incorrectly when it is their turn to move due to their neighbors on the other side of the target. Those at distance between $r$ and $2r$ from the separator might also update incorrectly (due to "corruption" from neighbors at distance $< r$ from the separator that had earlier updated incorrectly) but we will show that with high probability this only happens in the last $1/4$ of the ordering. I.e., within the first $3N/4$ updates, with high probability there are no incorrect updates by sensors at distance between $r$ and $2r$ from the target. Finally, we show that with high probability, those at

distance greater than $2r$ *never* update incorrectly. This last part of the argument follows from two facts: (1) with high probability all such points begin with more correctly-labeled neighbors than incorrectly-labeled neighbors (so they will update correctly so long as no neighbors have previously updated incorrectly), and (2) after $3N/4$ total updates have been made, with high probability more than half of the neighbors of each such point have already (correctly) updated, and so those points will now update correctly no matter what their remaining neighbors do. Our argument for the sensors at distance in $[r, 2r]$ requires $r$ to be small compared to $(\frac{1}{2} - \eta)/\sqrt{d}$, and the final error is $O(r\sqrt{d})$, so the conclusion is we have a total error less than $\epsilon$ for $r < c\min[\frac{1}{2} - \eta, \epsilon]/\sqrt{d}$ for some absolute constant $c$.

We begin with a key lemma. For any given sensor, define its inside-neighbors to be its neighbors in the direction of the target separator and its outside-neighbors to be its neighbors away from the target separator. Also, let $\gamma = 1/2 - \eta$.

**Lemma 2** *For any $c_1, c_2 > 0$ there exist $c_3, c_4 > 0$ such that for $r \leq \frac{\gamma}{c_3 \sqrt{d}}$ and $N \geq \frac{c_4}{(r/2)^d \gamma^2} \ln(\frac{1}{r^d \gamma \delta})$, with probability $1 - \delta$, each sensor $x$ at distance between $r$ and $2r$ from the target separator has $m_x \geq \frac{c_1}{\gamma^2} \ln(4N/\delta)$ neighbors, and furthermore the number of inside-neighbors of $x$ that move before $x$ is within $\pm\frac{\gamma}{c_2}m_x$ of the number of outside neighbors of $x$ that move before $x$.*

**Proof:** First, the guarantee on $m_x$ follows immediately from the fact that the probability mass of the ball around each sensor $x$ is at least $(r/2)^d$, so for appropriate $c_4$ the expected value of $m_x$ is at least $\max[8, \frac{2c_1}{\gamma^2}] \ln(4N/\delta)$, and then applying Hoeffding bounds [12, 7] and the union bound. Now, fix some sensor $x$ and let us first assume the ball of radius $r$ about $x$ does not cross the unit sphere. Because this is random-order dynamics, if $x$ is the $k$th sensor to move within its neighborhood, the $k - 1$ sensors that move earlier are each equally likely to be an inside-neighbor or an outside-neighbor. So the question reduces to: if we flip $k - 1 \leq m_x$ fair coins, what is the probability that the number of heads differs from the number of tails by more than $\frac{\gamma}{c_2}m_x$. For $m_x \geq 2(\frac{c_2}{\gamma})^2 \ln(4N/\delta)$, this is at most $\delta/(2N)$ by Hoeffding bounds. Now, if the ball of radius $r$ about $x$ does cross the unit sphere, then a random neighbor is slightly more likely to be an inside-neighbor than an outside-neighbor. However, because $x$ has distance at most $2r$ from the target separator, this difference in probabilities is only $O(r\sqrt{d})$, which is at most $\frac{\gamma}{2c_2}$ for appropriate choice of constant $c_3$.[1] So, the result follows by applying Hoeffding bounds to the $\frac{\gamma}{2c_2}$ gap that remains. ∎

**Theorem 4** *For some absolute constants $c_3, c_4$, for $r \leq \frac{\gamma}{c_3 \sqrt{d}}$ and $N \geq \frac{c_4}{(r/2)^d \gamma^2} \ln(\frac{1}{r^d \gamma \delta})$, in random order dynamics, with probability $1 - \delta$ all sensors at distance greater than $2r$ from the target separator update correctly.*

PROOF SKETCH: We begin by using Lemma 2 to argue that with high probability, no points at distance between $r$ and $2r$ from the separator update incorrectly within the first $3N/4$ updates (which immediately implies that all points at distance greater than $2r$ update correctly as well, since by Theorem 2, with high probability they begin with more correctly-labeled neighbors than incorrectly-labeled neighbors and their neighborhood only becomes more favorable). In particular, for any given such point, the concern is that some of its inside-neighbors may have previously updated incorrectly. However, we use two facts: (1) by Lemma 2, we can set $c_4$ so that with high probability the total contribution of neighbors that have already updated is at most $\frac{\gamma}{8}m_x$ in the incorrect direction (since the outside-neighbors will have updated correctly, by induction), and (2) by standard concentration

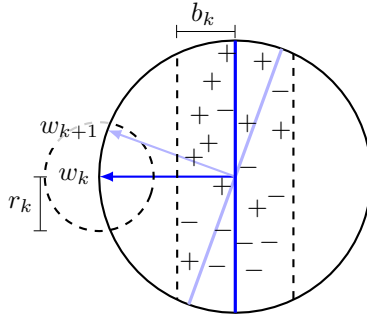

Figure 1: The margin-based active learning algorithm after iteration $k$. The algorithm samples points within margin $b_k$ of the current weight vector $w_k$ and then minimizes the hinge loss over this sample subject to the constraint that the new weight vector $w_{k+1}$ is within distance $r_k$ from $w_k$.

inequalities [12, 7], with high probability at least $\frac{1}{8}m_x$ neighbors of $x$ have *not* yet updated. These $\frac{1}{8}m_x$ un-updated neighbors together have in expectation a $\frac{\gamma}{4}m_x$ bias in the correct direction, and so with high probability have greater than a $\frac{\gamma}{8}m_x$ correct bias for sufficiently large $m_x$ (sufficiently large $c_1$ in Lemma 2). So, with high probability this overcomes the at most $\frac{\gamma}{8}m_x$ incorrect bias of neighbors that have already updated, and so the points will indeed update correctly as desired. Finally, we consider the points of distance $\geq 2r$. Within the first $\frac{3}{4}N$ updates, with high probability they will all update correctly as argued above. Now consider time $\frac{3}{4}N$. For each such point, in expectation $\frac{3}{4}$ of its neighbors have already updated, and with high probability, for all such points the fraction of neighbors that have updated is more than half. Since all neighbors have updated correctly so far, this means these points will have more correct neighbors than incorrect neighbors no matter what the remaining neighbors do, and so they will update correctly themselves. ∎

## 4   Query efficient polynomial time active learning algorithm

Recently, Awasthi et al. [1] gave the first polynomial-time active learning algorithm able to learn linear separators to error $\epsilon$ over the uniform distribution in the presence of agnostic noise of rate $O(\epsilon)$. Moreover, the algorithm does so with optimal query complexity of $O(d \log 1/\epsilon)$. This algorithm is ideally suited to our setting because (a) the sensors are uniformly distributed, and (b) the result of best response dynamics is noise that is low but potentially highly coupled (hence, fitting the low-noise agnostic model). In our experiments (Section 5) we show that indeed this algorithm when combined with best-response dynamics achieves low error from a small number of queries, outperforming active and passive learning algorithms without the best-response denoising step, as well as outperforming passive learning algorithms with denoising.

Here, we briefly describe the algorithm of [1] and the intuition behind it. At high level, the algorithm proceeds through several rounds, in each performing the following operations (see also Figure 1):

**Instance space localization:** Request labels for a random sample of points within a band of width $b_k = O(2^{-k})$ around the boundary of the previous hypothesis $w_k$.

**Concept space localization:** Solve for hypothesis vector $w_{k+1}$ by minimizing hinge loss subject to the constraint that $w_{k+1}$ lie within a radius $r_k$ from $w_k$; that is, $||w_{k+1} - w_k|| \leq r_k$.

[1, 10, 16] show that by setting the parameters appropriately (in particular, $b_k = \Theta(1/2^k)$ and $r_k = \Theta(1/2^k)$), the algorithm will achieve error $\epsilon$ using only $k = O(\log 1/\epsilon)$ rounds, with $O(d)$ label requests per round. In particular, a key idea of their analysis is to decompose, in round $k$, the error of a candidate classifier $w$ as its error outside margin $b_k$ of the current separator plus its error inside margin $b_k$, and to prove that for these parameters, a small constant error inside the margin suffices to reduce overall error by a constant factor. A second key part is that by constraining the search for $w_{k+1}$ to vectors within a ball of radius $r_k$ about $w_k$, they show that hinge-loss acts as a sufficiently faithful proxy for 0-1 loss.

# 5 Experiments

In our experiments we seek to determine whether our overall algorithm of best-response dynamics combined with active learning is effective at denoising the sensors and learning the target boundary. The experiments were run on synthetic data, and compared active and passive learning (with Support Vector Machines) both pre- and post-denoising.

**Synthetic data.** The $N$ sensor locations were generated from a uniform distribution over the unit ball in $\mathrm{R}^2$, and the target boundary was fixed as a randomly chosen linear separator through the origin. To simulate noisy scenarios, we corrupted the true sensor labels using two different methods: 1) flipping the sensor labels with probability $\eta$ and 2) flipping randomly chosen sensor labels and all their neighbors, to create pockets of noise, with $\eta$ fraction of total sensors corrupted.

**Denoising via best-response dynamics.** In the denoising phase of the experiments, the sensors applied the basic majority consensus dynamic. That is, each sensor was made to update its label to the majority label of its neighbors within distance $r$ from its location[2]. We used radius values $r \in \{0.025, 0.05, 0.1, 0.2\}$. Updates of sensor labels were carried out both through simultaneous updates to all the sensors in each iteration (synchronous updates) and updating one randomly chosen sensor in each iteration (asynchronous updates).

**Learning the target boundary.** After denoising the dataset, we employ the agnostic active learning algorithm of Awasthi et al. [1] described in Section 4 to decide which sensors to query and obtain a linear separator. We also extend the algorithm to the case of non-linear boundaries by implementing a kernelized version (see supplementary material for more details). Here we compare the resulting error (as measured against the "true" labels given by the target separator) against that obtained by training a SVM on a randomly selected labeled sample of the sensors of the same size as the number of queries used by the active algorithm. We also compare these post-denoising errors with those of the active algorithm and SVM trained on the sensors before denoising. For the active algorithm, we used parameters asymptotically matching those given in Awasthi et al [1] for a uniform distribution. For SVM, we chose for each experiment the regularization parameter that resulted in the best performance.

## 5.1 Results

Here we report the results for $N = 10000$ and $r = 0.1$. Results for experiments with other values of the parameters are included in the supplementary material. Every value reported is an average over 50 independent trials.

**Denoising effectiveness.** Figure 2 (left side) shows, for various initial noise rates, the fraction of sensors with incorrect labels after applying 100 rounds of synchronous denoising updates. In the random noise case, the final noise rate remains very small even for relatively high levels of initial noise. Pockets of noise appear to be more difficult to denoise. In this case, the final noise rate increases with initial noise rate, but is still nearly always smaller than the initial level of noise.

**Synchronous vs. asynchronous updates.** To compare synchronous and asynchronous updates we plot the noise rate as a function of the number of rounds of updates in Figure 2 (right side). As our theory suggests, both simultaneous updates and asynchronous updates can quickly converge to a low level of noise in the random noise setting (in fact, convergence happens quickly nearly every time). Neither update strategy achieves the same level of performance in the case of pockets of noise.

**Generalization error: pre- vs. post-denoising and active vs. passive.** We trained both active and passive learning algorithms on both pre- and post-denoised sensors at various label budgets, and measured the resulting generalization error (determined by the angle between the target and the learned separator). The results of these experiments are shown in Figure 3. Notice that, as expected, denoising helps significantly and on the denoised dataset the active algorithm achieves better generalization error than support vector machines at low label budgets. For example, at a

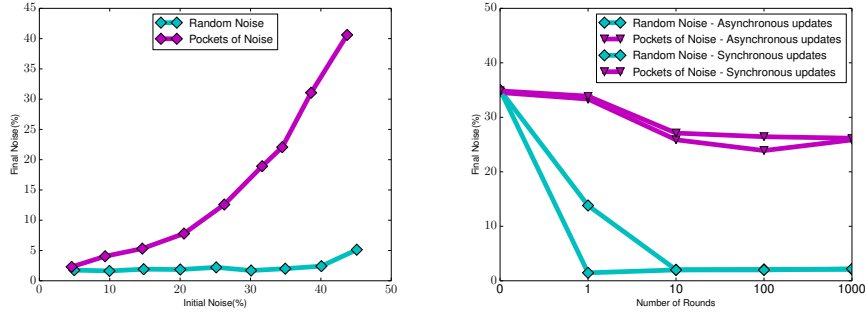

Figure 2: Initial vs. final noise rates for synchronous updates (left) and comparison of synchronous and asynchronous dynamics (right). One synchronous round updates every sensor once simultaneously, while one asynchronous round consists of $N$ random updates.

label budget of 30, active learning achieves generalization error approximately 33% lower than the generalization error of SVMs. Similar observations were also obtained upon comparing the kernelized versions of the two algorithms (see supplementary material).

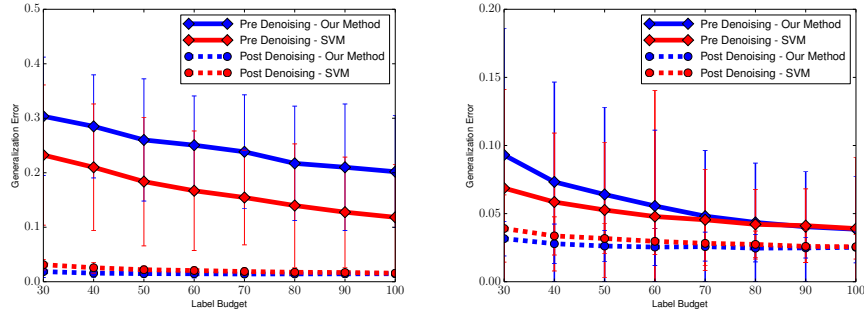

Figure 3: Generalization error of the two learning methods with random noise at rate $\eta = 0.35$ (left) and pockets of noise at rate $\eta = 0.15$ (right).

## 6 Discussion

We demonstrate through theoretical analysis as well as experiments on synthetic data that local best-response dynamics can significantly denoise a highly-noisy sensor network without destroying the underlying signal, allowing for fast learning from a small number of label queries. Our positive theoretical guarantees apply both to synchronous and random-order asynchronous updates, which is borne out in the experiments as well. Our negative result in Section 3.2 for adversarial-order dynamics, in which a left-to-right update order can cause the entire system to switch to a single label, raises the question whether an alternative dynamics could be robust to adversarial update orders. In the supplementary material we present an alternative dynamics that we prove is indeed robust to arbitrary update orders, but this dynamics is less practical because it requires substantially more computational power on the part of the sensors. It is an interesting question whether such general robustness can be achieved by a simple practicall update rule. Another open question is whether an alternative dynamics can achieve better denoising in the region near the decision boundary.

### Acknowledgments

This work was supported in part by NSF grants CCF-0953192, CCF-1101283, CCF-1116892, IIS-1065251, IIS1116886, NSF/NIH BIGDATA 1R01GM108341, NSF CAREER IIS1350983, AFOSR grant FA9550-09-1-0538, ONR grant N00014-09-1-0751, and Raytheon Faculty Fellowship.

## Footnotes

[1]We can analyze the difference in probabilities as follows. First, in the worst case, $x$ is at distance exactly $2r$ from the separator, and is right on the edge of the unit ball. So we can define our coordinate system to view $x$ as being at location $(2r, \sqrt{1 - 4r^2}, 0, \ldots, 0)$. Now, consider adding to $x$ a random offset $y$ in the $r$-ball. We want to look at the probability that $x + y$ has Euclidean length less than 1 conditioned on the first coordinate of $y$ being negative compared to this probability conditioned on the first coordinate of $y$ being positive. Notice that because the second coordinate of $x$ is nearly 1, if $y_2 \leq -cr^2$ for appropriate $c$ then $x + y$ has length less than 1 no matter what the other coordinates of $y$ are (worst-case is if $y_1 = r$ but even that adds at most $O(r^2)$ to the squared-length). On the other hand, if $y_2 \geq cr^2$ then $x + y$ has length greater than 1 also no matter what the other coordinates of $y$ are. So, it is only in between that the value of $y_1$ matters. But notice that the distribution over $y_2$ has maximum density $O(\sqrt{d}/r)$. So, with probability nearly $1/2$, the point is inside the unit ball for sure, with probability nearly $1/2$ the point is outside the unit ball for sure, and only with probability $O(r^2\sqrt{d}/r) = O(r\sqrt{d})$ does the $y_1$ coordinate make any difference at all.

[2]We also tested distance-weighted majority and randomized majority dynamics and experimentally observed similar results to those of the basic majority dynamic.

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
