[Supplementary Material]

# Active Learning and Best-Response Dynamics: Supplementary Material

**Maria-Florina Balcan**
Carnegie Mellon
ninamf@cs.cmu.edu

**Christopher Berlind**
Georgia Tech
cberlind@gatech.edu

**Avrim Blum**
Carnegie Mellon
avrim@cs.cmu.edu

**Emma Cohen**
Georgia Tech
ecohen@gatech.edu

**Kaushik Patnaik**
Georgia Tech
kpatnaik3@gatech.edu

**Le Song**
Georgia Tech
lsong@cc.gatech.edu

## A   Arbitrary Order and Conservative Best Response Dynamics

Given the negative result of Section 3.2, the basic best-response dynamics would not be appropriate to use if no assumptions can be made about the order in which sensors perform their updates. To address this problem, we describe here a modified dynamics that we call *conservative* best-response. The idea of this dynamics is that sensors only change their state when they are confident that they are not on the wrong side of the target separator. This dynamics is not as practical as regular best-response dynamics because it requires substantially more computation on the part of the sensors. Nonetheless, it demonstrates that positive results for arbitrary update orders are indeed achievable.

**Conservative best-response dynamics:**   In this dynamics, sensors behave as follows:

1. If, for all linear separators through the sensor's location, a majority of neighbors on both sides of the separator are positive, then flip to positive.
2. If, for all linear separators through the sensor's location, a majority of neighbors on both sides of the separator are negative, then flip to negative.
3. Else (for some linear separator through the sensor's location, the majority on one side is positive and the majority on the other side is negative) then don't change.
4. To address sensors near the boundary of the unit sphere, in (1)-(3) we only consider linear separators with at least $1/4$ of the points in their neighborhood on each side.

**Theorem 5** *For absolute constants $c_3, c_4$, for $r \leq \frac{\gamma}{c_3 \sqrt{d}}$ and $N \geq \frac{c_4}{(r/2)^d \gamma^2} \ln(\frac{1}{r^d \gamma \delta})$, in arbitrary-order conservative best-response dynamics, each sensor whose $r$-ball does not intersect the target separator will flip state correctly and no sensor will ever flip state incorrectly.*

Thus, Theorem 5 contrasts nicely with the negative result in Theorem 3 for standard best-response dynamics and shows that the potential difficulties of arbitrary-order dynamics no longer apply.

**Proof:**   We will show that for the given value of $N$, with high probability we have the following initial conditions: for each of the $N$ sensors, for all hemispheres of radius $r$ centered at that sensor, the empirical fraction of points in that hemisphere that are labeled positive is within a $\gamma = 1/2 - \eta$ fraction of its expectation. This implies in particular that at the start of the dynamics, all such hemispheres that are fully on the positive side of the target separator have more positive-labeled sensors than negative-labeled sensors, and all such hemispheres that are fully on the negative side of the target separator have more negative-labeled sensors than positive-labeled sensors. This in turn implies, by induction, that in the course of the dynamics, no sensor *ever* switches to the incorrect label. In particular, if we consider the hyperplane through a sensor that is parallel to the target and consider the hemisphere of neighbors on the "good side" of this hyperplane, by induction none of those neighbors will have ever switched to the incorrect label, and so their majority will remain the

correct label, and so the current sensor will not switch incorrectly by definition of the dynamics. In addition, it implies that all sensors whose $r$-balls do not intersect the target separator *will* flip to the correct label when it is their turn to update.

To argue this, for any fixed sensor and its neighborhood $r$-ball, since the VC-dimension of linear separators in $R^d$ is $d + 1$, so long as we see

$$m \geq \frac{c}{\gamma^2} \left[ d \ln(1/\gamma) + \ln(N/\delta) \right]$$

points inside that $r$-ball (for sufficiently large constant $c$), with probability at least $1 - \delta/N$, for any hyperplane through the center of the ball, the number of positives in each halfspace will be within $\gamma m/8$ of the expectation, and the number of negatives in each halfspace will be within $\gamma m/8$ of the expectation. This means that if the halfspace is fully on the positive side of the target, then we see more positives than negatives, and if it is fully on the negative side of the target then we see more negatives than positives. (In the case of sensors near the boundary of the unit ball, this holds true for all with sufficiently many points, which includes the halfspace defined by the hyperplane parallel to the target separator if the sensor is within distance $r$ of the target separator for $r < \frac{\gamma}{c_3 \sqrt{d}}$.) We are using $\delta/N$ as the failure probability so we can do a union bound over all the $N$ balls. Finally, we solve for $N$ to ensure the above guarantee on $m$ to get

$$N \geq \frac{c_4}{(r/2)^d \gamma^2} \ln \left( \frac{1}{r^d \gamma \delta} \right)$$

points suffice for some constant $c_4$. ∎

## B    Additional Proofs

**Proof (Theorem 3):** Suppose the labeling is given by $\text{sign}(w \cdot x)$. We show that if sensors are updated in increasing order of $w \cdot x$ (from most negative to most positive) then with high probability all sensors will update to negative labels.

Consider what we see when we come to update the sensor at $x$. Assuming we have not yet failed (given a positive label), all of the points $x'$ with $w \cdot x' < w \cdot x$ are labeled negative, while those with $w \cdot x' > w \cdot x$ are unchanged from their original states, and so are still labeled with independent uniform noise. As in the proof of Theorem 2, we apply Bernstein's theorem to the difference $\Delta$ between the number of negative and positive points in the neighborhood of $x$, which we write as a sum of $(N - 1)$ independent variables $\Delta_i$. The expected labels of the nearby points depend on the location of $x$, so we consider three regions: $w \cdot x \leq -r$, $w \cdot x \geq 0$, and $-r < w \cdot x < 0$.

Let $V$ denote the probability mass of the ball of radius $r$ around $x$. In all cases the variance is bounded by $\text{Var}[\Delta_i] \leq \text{E}[\Delta_i^2] = V \leq r^d$.

In the first region ($w \cdot x \leq -r$) we can use the same analysis from Theorem 2 to find that $\text{E}[\Delta_i] \leq -V(1 - 2\eta) \leq -(r/2)^d (1 - 2\eta)$, since the ball around $x$ never crosses the separator and any sensors previously updated to negative labels cannot hurt.

In the second region ($w \cdot x \leq 0$) we can use a similar analysis, bounding

$$\text{E}[\Delta_i] \leq -V/2 + (1 - \eta)V/2 = -\eta V/2 \leq -\tfrac{1}{2}(r/2)^d,$$

since the measure of the (positive biased) half of the ball further from the separator than $x$ is never larger than the measure of the remaining (all negative) half of the ball.

In the final region ($0 < w \cdot x < r$), we must take a little more care, as the measure of the all-negative half of the ball may be less than the measure of the unexamined side, which may be positive-biased due to crossing the separator. To analyze this case, we project onto the 2-dimensional space spanned by $x$ and $w$. The worst case is clearly when $x$ is on the surface of the ball, as shown in Figure 4.

Any point in the red region is known to have a negative label, while points in the dark blue region are biased towards positive labels. We first show that the red region is bigger by showing that the angle $\alpha$ subtended by the dark blue region is smaller than the angle $\beta$ of the red region. Construct the segment $\overline{xA}$ by reflecting the segment $\overline{xB}$ about the line $\overline{xO}$ and extending it to the separator. Note that the angle $\angle OxA$ is the same as the angle $\theta$ between $x$ and the separator. We find that $\alpha \leq \beta$ precisely when $xA \geq xC = r$. Indeed, by considering the isosceles triangle $\triangle AxO$ we see

Figure 4: A ball around $x$ intersecting the decision boundary and the boundary of the unit ball.

that $xA = 1/(2\cos\theta) \geq 1/2$. So as long as $r \leq 1/2$ we have $\beta \geq \alpha$. Thus, since the projection of the uniform distribution over the unit ball onto this plane is radially symmetric, the red region has more probability mass than the blue region.

We can now calculate for this case

$$\mathrm{E}[\Delta_i] \leq (-1)[\text{measure of red}] + (1-2\eta)[\text{measure of blue}] + (2\eta-1)[\text{measure of white}]$$
$$\leq -2\eta[\text{measure of red}].$$

Note that although the projection does not make sense for $d = 1$ the result obviously still holds (as there are no points near both the separator and the boundary of the unit ball). We can lower bound the measure of the red region by the measure of the sphere inscribed in the sector, which has radius at least $cr$ for some $0 < c < 1/2$ as long as $r \leq 1/2$ (since $\beta$ is bounded away from 0 in this range of $r$).

Now we see that for any $x$ the expected label satisfies

$$\mathrm{E}[\Delta_i] \leq -\tfrac{1}{2}(cr)^d \min(\eta, \tfrac{1}{2} - \eta).$$

Letting $\phi = \min(\eta, \tfrac{1}{2} - \eta)$, we find that the probability of giving a positive label on any given update is

$$\mathbb{P}[\Delta \geq 0] \leq \exp\left(\frac{-\tfrac{1}{4}(N-1)^2(cr)^{2d}\phi^2/2}{(N-1)r^d + (N-1)(cr)^d\phi/3}\right)$$
$$= \exp\left(\frac{-\tfrac{1}{4}(N-1)(cr)^d\phi^2}{1 + \phi/3}\right)$$
$$= \exp\left(-(N-1)(cr)^d\phi^2/8\right)$$

By the union bound, we find that

$$N \geq \frac{16}{(cr)^d\phi^2}\left(\ln\frac{8}{(cr)^d\phi^2} + \ln\frac{1}{\delta}\right)$$

suffices to ensure that with probability at least $1 - \delta$ all sensors are updated to negative labels. ∎

**Note 1** *If $r = O(1/\sqrt{d})$ then we can lower bound all of the relevant measures in the preceding proof by $\Theta(r^d)$ rather than $(\Theta(r))^d$, to see that*

$$N \geq \Omega\left(\frac{1}{r^d\phi^2}\left(\ln\frac{1}{r\phi} + \ln\frac{1}{\delta}\right)\right)$$

*suffices.*

Figure 5: Generalization error with 1000 sensors in different noise scenarios. Generalization error in y-axis and Labels used in x-axis. From Left to Right - Random Noise, and Pockets of Noise.

Figure 6: Generalization error with 5000 sensors in different noise scenarios. Generalization error in y-axis and Labels used in x-axis. From Left to Right - Random Noise, and Pockets of Noise.

# C  Additional Experimental Results

All of the following experiments were run with initial noise rate $\eta = 0.35$ for random noise and $\eta = 0.15$ for pockets of noise, and the results have been averaged over 20 trials of 50 iterations each.

**Effect of number of sensors on denoising and learning.**

We analyze the performance of learning post-denoising as a function of the number of sensors for a fixed radius. Given the results of Theorem 2 in Section 3.1 for synchronous updates, we expect the denoising to improve as sensors are added, which in turn should improve the generalization error post-denoising. Figures 5, 6, and 7 show the generalization error pre- and post-denoising for $N \in \{1000, 5000, 25000\}$. For a budget of 30 labels on random noise, the noise rate after denoising drops from 12.0% with 1000 sensors to 1.7% with 25000 sensors, and with this improvement we see a corresponding drop in generalization error from 7.4% to 1.6%. Notice that denoising helps for both active and passive learning in all scenarios except for the case of pockets of noise with 1000 sensors, where the sensor density is insufficient for the denoising process to have significant effect.

**Effect of communication radius on denoising and learning.**

We analyze here the performance of learning post-denoising as a function of the communication radius for a fixed number of sensors. In light of Theorem 2 we expect a larger communication radius to improve the effectiveness of denoising. Figures 8, 9, and 10 show the generalization error pre- and post-denoising for $r \in \{0.2, 0.05, 0.025\}$ with 10000 sensors. Here denoising helps for both active and passive learning in all scenarios.

Figure 7: Generalization error with 25000 sensors in different noise scenarios. Generalization error in y-axis and Labels used in x-axis. From Left to Right - Random Noise, and Pockets of Noise.

Figure 8: Generalization error with connectivity radius of 0.2 and 10,000 sensors in different noise scenarios. Generalization error in y-axis and Labels used in x-axis. From Left to Right - Random Noise, and Pockets of Noise.

Figure 9: Generalization error with connectivity radius of 0.05 and 10,000 sensors in different noise scenarios. Generalization error in y-axis and Labels used in x-axis. From Left to Right - Random Noise, and Pockets of Noise.

Figure 10: Generalization error with connectivity radius of 0.025 and 10,000 sensors in different noise scenarios. Generalization error in y-axis and Labels used in x-axis. From Left to Right - Random Noise, and Pockets of Noise.

## C.1 Kernelized Algorithm Derivation and Results

**Derivation of dual with a linear ball constraint**

In order to be able to replace inner products with kernel evaluations in the dual program of the hinge loss minimization step, we replace the ball constraint given by $\|w_k - w_{k+1}\|_2 \leq r_k$ with an equivalent linear constraint $w_k w_{k-1} \geq 1 - r_k^2/2$.

$$L = \sum_{i=1}^{n} \xi_i - \sum_{i=1}^{n} \alpha_i(y_i(w_k x_i)/\tau_k - 1 + \xi_i)$$
$$+ \beta(1 - r_k^2/2 - w_k w_{k-1})$$
$$+ \gamma(\|w_k\|_2 - 1) - \sum_{i=1}^{n} \delta_i \xi_i$$

To obtain the dual formulation, we take derivation of the above equation w.r.t to $w_k$ and $\xi$ and substitute these values in the original formulation we obtain

$$\max_{\alpha,\beta,\gamma} \sum_{i=1}^{n} \alpha_i - 1/4\gamma\tau_k^2 * \sum_{i=1}^{n}\sum_{j=1}^{n} \alpha_i\alpha_j y_i y_j x_i x_j$$
$$- \beta w_{k-1}/2\tau_k\gamma * \sum_{i=1}^{n} \alpha_i y_i x_i$$
$$- \beta^2 w_{k-1}^2/4\gamma + \beta(1 - r_k^2/2) - \gamma \tag{1}$$
$$\text{s.t}$$
$$0 \leq \alpha \leq 1$$
$$\beta, \gamma \geq 0$$

In (1) the lagrangian variable $\gamma$ is present as a denominator in three terms which are negative and as a quantity being subtracted in the objective function. Thus the edge values $0, \inf$ of $\gamma$ would lead to decrease in the objective value and cannot lead to maximum. Thus the maximum value of objective function will be found at $\gamma$ evaluated at $\partial L/\partial \gamma = 0$. Taking the derivative of 1 w.r.t $\gamma$ we get

$$\gamma = \sqrt{1/\tau_k^2 \sum_{i=1}^{n}\sum_{j=1}^{n} \alpha_i\alpha_j y_i y_j x_i x_j + \beta w_{k-1}/2\tau_k + \beta^2 w_{k-1}^2/4} \tag{2}$$

Figure 11: Accuracy with Gaussian kernels in different noise scenarios. Accuracy in y-axis and Labels used in x-axis. From Left to Right - Random Noise, and Pockets of Noise.

Substituting this value in the (1) and simplifying gives us -

$$\min_{\alpha,\beta} \sqrt{\sum_{i=1}^{n}\sum_{j=1}^{n} \alpha_i \alpha_j y_i y_j x_i x_j + 2\tau_k \beta w_{k-1}(\sum_{i=1}^{n} \alpha_i y_i x_i) + (\beta \tau_k w_{k-1})^2}$$
$$-\tau_k \sum_{i=1}^{n} \alpha_i - \tau_k \beta (1 - r_k^2/2) \qquad (3)$$
$$\text{s.t}$$
$$0 \le \alpha \le 1$$
$$\beta \ge 0$$

The term under the square root in (3) can be simplified as $(\sum_{i=1}^{n} \alpha_i y_i x_i + \beta \tau_k w_{k-1})^2$, which further simplifies equation (3) to

$$\max_{\alpha,\beta} \tau_k \sum_{i=1}^{n} \alpha_i + \tau_k \beta (1 - r_k^2/2)$$
$$- \left\| \sum_{i=1}^{n} \alpha_i y_i x_i + \beta \tau_k w_{k-1} \right\| \qquad (4)$$
$$\text{s.t}$$
$$0 \le \alpha \le 1$$
$$\beta \ge 0$$

The resulting optimization objective function can be implemented by expanding out the two norm and value of previous weight vector as $\sum_{l=1}^{p} \alpha_l y_l x_l$.

**Results for kernelized algorithm**

We also test the improvement of the active learning method for non-linear decision boundaries. The target decision boundary is a sine curve on the horizontal axis in $R^2$ space, with points above the curve labeled as positive, else negative. Noise was introduced in the true labels through methods described in Section 5.1. For comparison with passive methods we calculate the classification error over 20 trials, where in each trial we average results over 20 iterations. Both the active and passive algorithms use a Gaussian kernel with bandwidth of 0.1 for a smooth estimate of the the boundary. All other parameters remain the same. Results are shown in Figures 6. Notice that the results are similar to experiments with linear decision boundaries.