[Reviews · NeurIPS 2014]

Submitted by Assigned_Reviewer_11

The paper considers the setting of a sensor network (or agents) in a noisy environment that are able to communicate locally. The authors prove that theoretical bounds on the number of active queries can be achieved through simple best response dynamics.

The paper is very well-written, technically correct, and the synthetic experiment makes the results clear. The theoretical results are novel and I think that the paper deserves to be published to be published at NIPS.

For the benefit of readers, the paper needs to cite, and relate to, recent papers in NIPS that have presented theoretical guarantees for active learning in the presence of noise:
1. Near-optimal Bayesian active learning with noisy observations. Golovin et al. NIPS 2010.
2. Extensions of generalized binary search to group identification and exponential costs. Bellala et al. NIPS 2010.
Summary: This paper presents novel and interesting theoretical results for a sensor network in equilibrium of a consensus game, obtained through simple best response dynamics, in a noisy environment.

Submitted by Assigned_Reviewer_28

SUMMARY:
This paper combines dynamics from game theory and active learning techniques to solve a problem where a center tries to determine a decision boundary from noisy sensors. Under the assumption that the noise is independent, the authors prove that simple simultaneous or asynchronous game dynamics successfully denoise the system with high probability, given enough sensors. To illustrate the effectiveness of the denoising together with active learning (which is known to perform poorly in the presence of large amounts of noise), the authors give experiments in the plane which show superior results compared with learning without denoising and passive learning with denoising.

CRITIQUE:
The results are interesting, the approach is novel, and the exposition is clear. My only concern is the significance. I believe the authors make a good case in general that denoising a collection of sensors, using simple distributed local computations, can be effective enough in some situations to allow active learning to be applied, thereby enabling an efficient sensor query load. However, the bounds given in the analysis leave a lot of room to question the applicability of this approach in practice. Is it really reasonable to have N = 10000 for r = 0.1, as in the experiments? Some rough sense of reasonable settings of these parameters is badly needed to evaluate the utility of this approach. For example, if r = 0.8, then surely one would want a more sophisticated denoising algorithm, since the theoretical guarantees break down, and moreover nodes have unreasonable communication loads. Similarly, if r=0.001, then the communication graph is likely to be disconnected, unless N is even larger. What settings of r, N, eta, etc are reasonable? Likewise, is a uniform distribution reasonable?

As another high level point, I was surprised not to see more discussion of the 2r band around the separator within which all bets are off -- are we assuming r is so small that this is negligible? For large r, it seems that a slightly more sophisticated distance-weighted approach would be much better here. This (or some similar procedure) is stated in a footnote to perform similarly to the simple majority; intuitively, the two should behave roughly the same away from the boundary, but the distance weighting should perform significantly better near the boundary, even for smaller r. A more fine-grained comparison near the boundary would be nice here.

SPECIFIC COMMENTS:
pg 1
. This paper would be much more accessible if terms such as active learning and agnostic active learning were defined in simple terms after their first use (e.g. "active learning, a branch of machine learning where...")
pg 2
. "sensors within distance r are connected by an edge" -- as mentioned above, the relationship between r and d (dim) and N is not discussed much in the text; aside from what values one should expect in practice, it might be good to mention some results from theoretical computer science / graph theory about the connectedness of the communication graph.
pg 3
. it would help to give an overview of section 3 at the top, since it was not stated prior
pg 7
. "Pockets of noise appear to be more difficult to denoise." -- the algorithm was not designed for this case, so this is not surprising. One might expect pockets to be less severe in higher dimensions.
pg 8
. "A synchronous round" -- perhaps change to "One synchronous round" to avoid confusion with "asynchronous"
Summary: Combines consensus dynamics and active learning techniques to solve a problem where a center tries to determine a decision boundary from noisy sensors. Clear, interesting, novel, but needs more justification re significance.

Submitted by Assigned_Reviewer_33

The authors study the following problem. There are N noisy sensors with local communication capabilities uniformly distributed in a region. There is an underlying function separating the region into positive and negative sub-regions. The objective of the external source is to accurately separate the region with as few queries from the source to sensors as possible.

The main idea is that local communication amongst sensors can be used to de-noise the system before costly communication with the external source. The authors analyze versions of a best-response dynamic that asynchronously or synchronously updates a sensor’s reading to the majority reading of its neighbors. They combine the de-noising with an active learning algorithm from the literature, and show through simulations that this combination outperforms SVMs with de-noising as well as either algorithm prior to de-noising.

This work is interesting and relevant to NIPS, and is very well-presented. The technical results (both analytical and simulations) are of high quality.

Some minor issues I had:

— Figure 3 should have y-axes on the same scale so that random noise and pockets of noise can more easily be compared.

— Based on the simulations, almost all the benefit is coming from de-noising the data. After the data is de-noised, SVM and active learning perform about the same (there are no error bars on the plots, but the difference does not appear to be meaningful regardless of significance). The authors call active learning “our method”, but the real contribution seems to be the de-noising. Hopefully this can be clarified in the paper.

— De-noising performance was much better on random noise than pockets of noise in simulations, but the pockets of noise model (or generally, noise that is not independent across sensors) seems more realistic. This result somewhat weakens the contribution of the de-noising algorithm.

— In Figure 2 (right), for synchronous updates with random noise, it looks like there is a slight increase in final noise as the number of rounds increases. This made me wonder: how do results look for 100K rounds with more simulations, and what does the max over final noise look like after each number of rounds? Given that the analytical results were for a single step with independent noise, and that running multiple steps will violate the independent noise assumption, it seems that there could be instances where error in an early step propagates to result in high final noise.

— Not really an issue with the paper, but for consideration: the motivation of the de-noising algorithm is that local communication is cheap while communication with the external source is expensive. Practically speaking, another alternative would be for the sensors to communicate locally to collect information (without the best-response dynamic), and then have a single message sent to the source with information about all sensor readings. If this approach is impractical for some reason (e.g., it still requires more data to be sent to the external source, albeit by a single sensor), it might be worth further specifying the problem to rule this possibility out.
Summary: A well-presented paper with interesting analytical and simulation-based results.
Author Feedback
Author rebuttal: We thank all the reviewers for their thoughtful comments. We are indeed excited by this new combination of active learning and game-theoretic dynamics for engineered systems. As reviewers point out, a key contribution from the perspective of active learning is to show that simple game-theoretic dynamics can bring a natural system from a high-noise state to the low-noise regime where active learning can be usefully applied. From the perspective of game theory, a key contribution is to analyze not just convergence to some equilibrium, but rather the quality of the states produced with respect to the overall system goals. Below are answers to specific points and questions of the reviewers:

Reviewer 1: Thanks. We will be sure to cite and relate to the papers you mentioned in our final version.

Reviewer 2:
- Regarding the values of N and r: our motivation is the setting of cheap low-power sensors, so we are viewing r as small, and asking how large must N be for this approach to help. Note that at the very least N must be large enough for the radius-r balls around sensors to be non-empty in order for communication to even be possible, and our theoretical results kick in when N is not *too* much larger than that. If r is much larger than needed for the given number of sensors N, then indeed one would want to just use signals from nearby neighbors, i.e., to effectively “shrink” r down to the smallest possible value given N. But since our motivation is small r we decided not to put this into the dynamics. Regarding the uniform distribution: since the sensors are put in place by the monitor, the monitor can choose to distribute them uniformly (which is natural to do with sensors). So we believe the uniform distribution is quite reasonable.

- It is true that one might hope that a more sophisticated procedure might be able to further improve performance near the boundary. Indeed, we experimented with distance-weighted de-noising but found that this did not help significantly (footnote 2). The problem is that (in either approach) the de-noised boundary becomes smooth but just not completely linear. We can add additional discussion of this in the final version.

- Thank you for your suggestions on the write-up. We will incorporate them in the final version as well.

Reviewer 3:
- Regarding the simulations, de-noising indeed provides most of the benefit, though at the low label budgets, active learning gives an additional 33% improvement. We will add error bars and clarify.

- Regarding random noise versus pockets of noise: the pockets-of-noise model was intended to be a worst-case scenario for the de-noising procedure (since the de-noising is based on just local interactions). We view it as a positive sign that even in this difficult case there is a noticeable gain.

- Your question about multiple rounds is a good one. In simulation, we observe that nearly all of the time (the curve shown is an average over multiple runs) after very few rounds the system reaches equilibrium and no more state-changes occur. We will add additional discussion of this point in the final version.

- Your last comment about the model raises an interesting point. However, even with this capability, unless the sensor can also record exactly which neighbor gave which reading, it's unclear whether sending the entire batch of all neighboring sensor readings would give substantial benefit. It also would add significant complexity to the sensors themselves (which would go against the motivation of cheap sensors) and to the message lengths. It is an interesting question, though, whether other simple types of information could yield an improvement.